# Dynamic Assembly/Disassembly of *Staphylococcus aureus* FtsZ Visualized by High-Speed Atomic Force Microscopy

**DOI:** 10.3390/ijms22041697

**Published:** 2021-02-08

**Authors:** Junso Fujita, Shogo Sugiyama, Haruna Terakado, Maho Miyazaki, Mayuki Ozawa, Nanami Ueda, Natsuko Kuroda, Shun-ichi Tanaka, Takuya Yoshizawa, Takayuki Uchihashi, Hiroyoshi Matsumura

**Affiliations:** 1Graduate School of Frontier Biosciences, Osaka University, 1-3 Yamadaoka, Suita, Osaka 565-0871, Japan; jfujita@fbs.osaka-u.ac.jp; 2Department of Physics, Nagoya University, Furo-cho, Chikusa-ku, Nagoya, Aichi 464-8602, Japan; sugiyama.0270@gmail.com; 3Department of Biotechnology, College of Life Sciences, Ritsumeikan University, 1-1-1 Noji-higashi, Kusatsu, Shiga 525-8577, Japan; t.d.o.v.b.h.y68@icloud.com (H.T.); m.miyazaki.2017m2@gmail.com (M.M.); sb0052fi@ed.ritsumei.ac.jp (M.O.); n.ueda.2018m2@gmail.com (N.U.); sb0063ie@ed.ritsumei.ac.jp (N.K.); stanaka1@kpu.ac.jp (S.-i.T.); t-yosh@fc.ritsumei.ac.jp (T.Y.); 4Department of Biomolecular Chemistry, Kyoto Prefectural University, Hangi-cho, Shimogamo, Sakyo-ku, Kyoto 606-8522, Japan; 5Exploratory Research Center on Life and Living Systems (ExCELLS), National Institutes of Natural Sciences, 5-1 Higashiyama, Myodaiji, Okazaki 444-8787, Japan

**Keywords:** bacterial cell division, *Staphylococcus aureus*, FtsZ, high-speed atomic force microscopy

## Abstract

FtsZ is a key protein in bacterial cell division and is assembled into filamentous architectures. FtsZ filaments are thought to regulate bacterial cell division and have been investigated using many types of imaging techniques such as atomic force microscopy (AFM), but the time scale of the method was too long to trace the filament formation process. Development of high-speed AFM enables us to achieve sub-second time resolution and visualize the formation and dissociation process of FtsZ filaments. The analysis of the growth and dissociation rates of the C-terminal truncated FtsZ (FtsZt) filaments indicate the net growth and dissociation of FtsZt filaments in the growth and dissociation conditions, respectively. We also analyzed the curvatures of the full-length FtsZ (FtsZf) and FtsZt filaments, and the comparative analysis indicated the straight-shape preference of the FtsZt filaments than those of FtsZf. These findings provide insights into the fundamental dynamic behavior of FtsZ protofilaments and bacterial cell division.

## 1. Introduction

FtsZ is a tubulin-homolog GTPase protein that is highly conserved among bacterial and archaeal species [1,2,3,4]. During cell division, FtsZ gathers to form a filamentous shape called a “protofilament” in the presence of GTP [5]. Protofilaments further assemble into a ring (Z-ring) [6], which is tethered to the cytoplasmic membrane by anchor proteins, such as FtsA and ZipA through flexible C-terminal tail of FtsZ [7,8]. Thus, the Z-ring is tightly associated with the membrane and cell wall, where peptidoglycan synthesis and remodeling occur during cell shape change and septum formation [9]. Recent studies have shown that GTPase activity-coupled treadmilling of FtsZ protofilaments regulates peptidoglycan synthesis [10,11]; highlighting the importance of FtsZ dynamics in bacterial cell division, but the whole process is still not fully understood. To address this question, a large number of imaging studies for visualizing the protofilaments and Z-ring has been performed both in vivo and in vitro. Most commonly used are fluorescence microscopy [12,13,14], electron microscopy [15,16,17], electron tomography [18,19], and atomic force microscopy (AFM) [20,21,22,23]. Among these methods, AFM does not require labeled proteins and should be able to follow filament growth and dissociation by capturing images continuously. However, FtsZ filaments are highly dynamic and flexible: bending, circularization, fragmentation, annealing, and bundling have been observed by AFM [22,23], and these events seem to occur on a second time scale. Therefore, the limited time resolution of AFM (around a minute) has hampered understanding of the detailed mechanism. High-speed AFM (HS-AFM) developed by Dr. Toshio Ando and co-workers at Kanazawa University takes each image at a much faster frame rate (~1 frame s^−1^) [24,25] and can be a powerful tool for visualization of FtsZ polymer dynamics. They have already captured images of FtsZ protofilaments from *Escherichia coli* on a time scale of seconds [26]. However, the study mainly focused on the FtsZ modulator ClpX, and the detailed FtsZ dynamics were not investigated, likely because of the fast FtsZ polymer formation.

As the image resolution of AFM is not very high, other techniques such as X-ray crystallography and cryo-electron microscopy (cryo-EM) should be used complementarily to examine the detailed molecular mechanism of FtsZ. The GTPase domain of FtsZ can be further divided into three subdomains: N-terminal, C-terminal, and the long helix connecting the two subdomains [4]. Change of the relative subdomain arrangement affects the enzymatic activity and assembly/disassembly of FtsZ, because GTP binds between N- and C-terminal subdomains. Among bacterial species, many crystal structures of FtsZ from *Staphylococcus aureus* (SaFtsZ) are available, leading to structural insights [27,28]. Previously, we determined the crystal structures of both the tense (T) and relaxed (R) states of native SaFtsZ [29]. This study explained the assembly/disassembly mechanism of FtsZ, triggered by conformational changes induced by the relative movement between the N- and C-terminal subdomains, which had been proposed previously [30,31]. Recent progress of cryo-EM study provides a density map of FtsZ protofilaments from *Escherichia coli* [32], but the resolution is still too low to clarify detailed interactions.

Here, we visualized and analyzed the formation and dissociation of C-terminal truncated SaFtsZ (FtsZt) filaments using HS-AFM. We observed the complete dissociation of dense FtsZ filaments on the mica surface with addition of excess GDP on a time scale of seconds. Individual ends of the filaments showed random growth and dissociation on a short time scale, resulting in net growth or dissociation depending on the condition. We also analyzed the curvatures of the FtsZ filaments, and the comparative analysis indicated that a larger number of straight filaments were observed in FtsZt than in full-length FtsZ (FtsZf). These findings suggest an important role of the flexible C-terminal tail of FtsZ. The visualized assembly/disassembly process in this study helps to further clarify FtsZ dynamic features, as well as the entire cell division process.

## 2. Results

### 2.1. Observation of SaFtsZ Filaments with HS-AFM

In most of the previous FtsZ studies using AFM, the observation buffers contained high concentrations (500 mM) of potassium chloride (KCl) [20,21,22,23], presumably to decrease the strong interaction between FtsZ molecules and the mica surface. Therefore, we first tested various KCl concentrations, and all HS-AFM observations discussed were performed under the optimized pool buffer condition (50 mM Tris-HCl pH 7.5, 5 mM MgCl_2_, 100 mM KCl), and GTP or GDP were added after starting measurement. Notably, actual concentrations of free FtsZ and GTP/GDP in the solution cannot be determined because FtsZ molecules remain attached to the mica surface and GTP is hydrolyzed into GDP during the measurement. Thus, we describe the total added amount of GTP/GDP and the final GTP/GDP concentration. Additionally, as full-length SaFtsZ polymerizes very quickly, we used the truncated SaFtsZ construct containing only the GTPase enzymatic domain (residues 12–316; FtsZt) to investigate filament formation and dissociation. Although refolded FtsZt was used to minimize the effect of GDP bound during overexpression in *E. coli*, no clear difference was observed compared to non-refolded FtsZt.

Many FtsZ filaments were observed in the presence of 200 nM FtsZt and 600 μM GTP (Figure 1A), while no filaments were observed in the absence of GTP. We added much more GTP than FtsZ, because GTP is consumed during the process and high GTP concentration drives FtsZ to the T state and promotes polymerization. The height profile along a single filament (indicated by the red line in Figure 1A) shows the periodic structure (Figure 1B), and this period corresponds to that of the SaFtsZ filament (4.4 nm) in the previous crystal structure (Figure 1C, PDB entry: 3VOA) [28]. Therefore, we confirmed that these filaments are composed of SaFtsZ. However, we could not identify the orientation or conformation of each FtsZ monomer because of the lack of sufficient resolution.

### 2.2. Formation, Elongation, and Dissociation of SaFtsZ Filaments

Next, we investigated how FtsZ protofilaments form, grow, and dissociate by changing the concentrations of FtsZ and GTP/GDP in a stepwise manner. To slow down filament growth, we began with 100 nM FtsZt and 300 μM GTP. In this condition, FtsZ filaments formed and grew slowly, but did not cover all areas of the mica surface (Figure 2A and Appendix A). Most of the filaments were straight, or at least not severely bent. After a single filament was formed, another filament tended to elongate along the existing filament, possibly reflecting lateral interactions between FtsZ filaments. After GTP concentration was increased to 600 μM, the filaments extended longer, but no significant increase in filament number was observed, probably because fewer FtsZ molecules floated in the pool buffer (Figure 2B). Rapid growth of FtsZ filaments was observed after increasing the FtsZt concentration to 200 nM (Figure 2C and Appendix A). After incubation for 20 min, the area of the mica surface was mostly covered with straight filaments, arranged to align in one direction (Figure 2D). Then, the GDP concentration in the pool buffer was increased to 3 mM to investigate and promote dissociation of FtsZ filaments. We expected that a high GDP/GTP ratio might shift the equilibrium towards disassembly of FtsZ filaments. After 10 min of incubation, many gaps were generated between filaments, indicating the FtsZ filaments started dissociating slowly, although many filaments remained (Figure 2E). We also found that some gaps were filled with free FtsZ molecules floating in the pool buffer, but this reassembly was not as fast as the dissociation, resulting in a slow dissociation overall. After the GDP concentration was increased to 6 mM, the gaps rapidly expanded, and most of the filaments were completely dissociated after 25 min (Figure 2F and Appendix A). Thus, a high concentration of GDP dissociates FtsZ filaments attached to a mica surface. Notably, the filaments remained straight; bending of filaments was hardly observed, even when small numbers of filaments were left without lateral interaction.

To summarize and analyze these results, we measured the number and area of the filaments from each movie and plotted them over time (Figure 3). It should be noted that the time course is not completely continuous, as we had to add FtsZ or GTP/GDP and change the recording settings between each movie. However, we did not observe large changes in the number and length of filaments after recording the next movie because of slow diffusion of the additives. In the presence of 100 nM FtsZt and 300 μM GTP, both the number and area of filaments increased rapidly up to approximately 25 and 3000 pixels, respectively. When the GTP concentration was raised to 600 μM, the area fluctuated and finally reached 4500 pixels, although the number did not change significantly. This suggests that filament propagation rather than formation occurred in this step, probably because fewer free FtsZ monomers were present in the solution. After we added FtsZt to 200 nM, the number and area were increased very rapidly up to 80 and 14,000 pixels, respectively. Filament dissociation in the presence of 3 mM GDP was very slow due to the remaining filament growth. The number and area dropped suddenly, and the filaments were almost completely dissociated, after the addition of 6 mM GDP. Notably, the dissociation was still slow just after the addition of GDP, but suddenly accelerated after 5 min. This is probably because of the limited diffusion speed of additional GDP. Another possible explanation is that lateral interactions stabilize filament formation; therefore, it takes a long time to dissociate the filaments. Throughout these processes, FtsZ and GTP/GDP concentrations mainly contributed to the formation and growth/dissociation, respectively. These results demonstrate the power of high-speed AFM: this is the first visualization of the association and dissociation of FtsZ filaments on a second time scale, which is difficult to capture with non-high-speed AFM.

Next, we tried to estimate the growth and dissociation rates of FtsZt filaments. We picked several filaments from the images in the growth condition (200 nM FtsZt and 600 μM GTP) and the dissociation condition (200 nM FtsZt, 600 μM GTP, and 6 mM GDP.), respectively, and prepared kymographs to calculate growth and dissociation rates. There were some limitations on tracking single filaments. We had to pick up only straight and stable filaments, otherwise the edge of the filament could not be detected because of the bending, fission, and movement of the filament. Almost all edges of the filaments seem to repeat the elongation and dissociation randomly on a second time scale in the both conditions (Figure 4A,B). We estimated instantaneous growth and dissociation rates from the kymographs and plot them as histograms in each condition (Figure 4C,D). Again, both histograms seem to represent random motions of FtsZt filaments. In a certain moment, the filaments showed rapid propagation and dissociation (>20 nm/s) in the growth and dissociation condition, respectively. Although these rare events were not represented much in the histograms of a second time scale, they lead to net growth or dissociation of the filaments on a larger time scale.

### 2.3. Curvature of SaFtsZ Filaments

Then we observed the filaments of FtsZf under the conditions of 100 nM FtsZf and 600 μM GTP to compare with FtsZt. FtsZf filaments seemed to be more curved than those of FtsZt (Figure 5A and Appendix A), and we did not observe complete dissociation of FtsZf filaments even in the presence of 6 mM GDP (Figure 5B and Appendix A). Growth/dissociation rates were too difficult to calculate, because curvature of FtsZf filaments hampered tracking of single filaments and preparation of kymographs. Instead, to analyze filament curvature further, we picked 265 and 603 filaments from FtsZt and FtsZf in the same condition of 100 nM FtsZt/FtsZf and 600 μM GTP, respectively, and calculated radii of circles by regarding the filaments as arcs. The straighter filaments become, the larger the radius of the circle should be. Histograms of FtsZt and FtsZf are shown in Figure 5C,D, respectively. Both histograms showed similar shapes: a single peak with an extended tail. FtsZt showed wider distribution than FtsZf, and medians of the histograms were 262 and 114 nm in FtsZt and FtsZf, respectively. Larger median value in FtsZt reflects the straight-shape preference of the filaments.

## 3. Discussion

We observed the formation, growth, and dissociation of FtsZt filaments in the presence of sufficient GTP or GDP. Throughout these processes, most of the filaments remained straight or at least not severely curved or circularized. In contrast, many curved filaments were observed in FtsZf, corresponding with the previous observations [20,21,22,23]. The truncated region, flexible C-terminal tail of FtsZ, interacts not only with other cell division proteins, such as FtsA and ZipA [33,34,35], but also with another FtsZ molecule to stabilize the filaments [36]. Other previous studies showed that curvature and torsion of FtsZ filaments affected by the surface and the linker attached [21,37]. From these results, we speculated that intact FtsZ filaments tend to be curved and circularized on mica surface, and truncation of the flexible C-terminal tail leads to alter the interaction between FtsZ and mica surface, although the effect of truncation depends on the length and sequence of C-terminal tail and therefore on species.

Previous studies revealed a treadmilling behavior of FtsZ filaments in vivo [10,11], while such behavior has not been observed by AFM on mica surface [22,38]. For treadmilling, FtsZ filament needs to grow from one end and to shorten from the other end. It has been proposed that treadmilling of the filaments could be a consequence of attachment to the surface, which could explain different behaviors of the filaments between in a cell and in AFM [37]. For example, throughout the measurements, fission and parallel arrangement of the filaments were observed. These events were also reported in previous AFM measurements of FtsZ from *Escherichia coli* and described as fragmentation, annealing, and bundling [23]. Such behavior of FtsZ filaments may be one of the mechanisms for formation of several parallel FtsZ filaments and construction of Z-rings. However, we should keep in mind that we only can observe the molecular behavior on mica by this technique.

We investigated growth rates of many ends of the FtsZt filaments in the growth and dissociation conditions, and found that the motion on a second time scale appeared to be random rather than ordered, corresponding with previous observation [22]. The shorter time scale of HS-AFM enabled us to confirm the random behavior of both ends of the filaments, and rapid propagation or shortening at a specific moment mainly contributes net growth/dissociation on a minute time scale as previously observed. Note that we could analyze the growth and dissociation rates only for FtsZt, because of the limitations for tracking single filaments at the same point and preparing kymographs. Additionally, we did not find dissociation of severely-curved FtsZt filaments during the analysis. In other words, all filaments were disassembled in the presence of excess GDP without bending, even in the concentration where little lateral interactions left (Figure 2F and Appendix A). This behavior could be derived from truncation of the C-terminal tail, which affects dynamics of FtsZ on surfaces [39], but previous AFM study also observed an isolated single straight filament without lateral interactions [38]. As curved FtsZf filaments did not be packed as dense as those of FtsZt, and not completely dissociated even in the presence of excess GDP, it is tempting to assume that the flexible C-terminal tail of FtsZ stabilizes longitudinal interactions and the filaments with some flexibility of curvature. Further structural studies using cryo-EM as well as X-ray crystallography would help understanding the relationship of the dynamics/structure/function of FtsZ.

## 4. Materials and Methods

### 4.1. Cloning, Protein Expression, and Purification

Refolded SaFtsZt (residues 12–316) was cloned, overexpressed, and purified as we described previously [29], except a 5 mL HiTrapQ HP column (Cytiva, Marlborough, MA, USA) was used instead of a 1 mL Resource Q column and the following additional denaturing and refolding steps inserted after the anion exchange chromatography: the FtsZ fraction was dialyzed against denaturing buffer (50 mM Tris-HCl pH 7.5, 200 mM NaCl, 10% v/v glycerol, 6 M urea) for 2–3 h three times. The denatured sample was refolded by a dialysis against 50 mM Tris-HCl pH 7.5, 10% v/v glycerol for 3 h. The refolded FtsZ was purified again using 5 mL HiTrapQ HP column, and the protein was eluted by 30–750 mM NaCl gradient. We also tested non-refolded SaFtsZt, but no clearly different feature was observed compared to the refolded one. FtsZf (full-length) was overexpressed and purified as FtsZt.

### 4.2. High-Speed AFM

All AFM images were captured by a laboratory-built high-speed AFM apparatus in the tapping mode [24,25]. The cantilever (Olympus, Tokyo, Japan) shows a resonant frequency of ~1 MHz in water and a spring constant of ~0.16 N m^−1^. The laser, whose output is ~0.8 mW and wavelength is 680 nm, was focused onto the back side of the cantilever, and the reflected laser deflection from the cantilever was detected with an optical beam deflection detector. To obtain higher resolution images, an amorphous carbon tip was constructed on the original AFM cantilever by electron beam deposition [24,40]. The observation was started on a freshly cleaved mica surface under a 70–80 μL of pool buffer (50 mM tris-HCl pH 7.5, 5 mM MgCl_2_, 100 mM KCl), and purified SaFtsZ, GTP, and GDP were added, if required, during the measurements. High concentration stock solutions of SaFtsZ, GTP, and GDP were used to avoid dilution of the measuring pool buffer. All measurements were performed at room temperature.

### 4.3. Analysis of High-Speed AFM Data

To estimate the growth and dissociation rates of FtsZt filaments, we first constructed kymographs as shown in Figure 4A,B using a laboratory-developed AFM image viewer based on Igor Pro (Ver. 8.0.4, Wave-Metrics Inc., Lake Oswego, OR, USA). The filament-end positions of each frame in the kymograph were detected using the edge detection function of MatrixFiler, which is implemented in IgorPro. Then a graph of filament end position vs. time was made, and the growth and dissociation rates were estimated from the slope of the graph (Figure 4A,B).

To analyze filament curvature, we picked the filaments from FtsZt and FtsZf, respectively. The radii of circles are calculated by regarding the filaments as arcs on Igor Pro (Ver. 8.0.4, Wave-Metrics Inc., Lake Oswego, OR, USA)). Obtained radii values were analyzed by Igor Pro, giving the median values of the histograms.

## Figures and Tables

**Figure 1 ijms-22-01697-f001:**
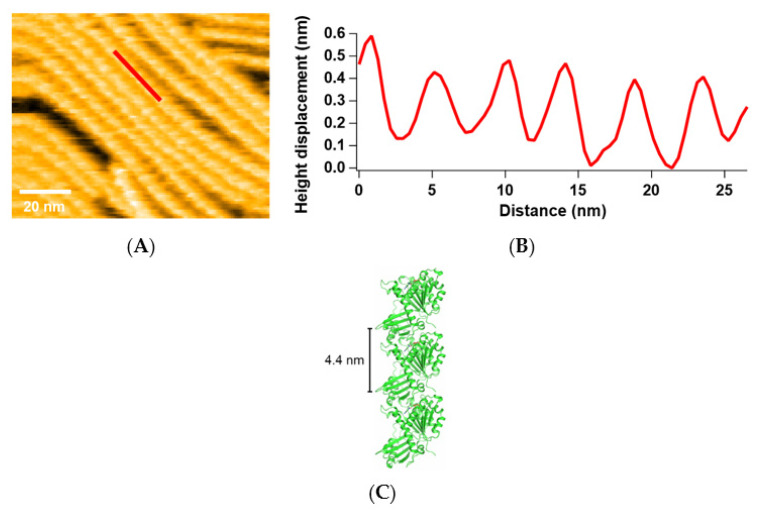
SaFtsZ filaments observed by HS-AFM: (**A**) An image of FtsZt filaments in the presence of 200 nM FtsZt and 600 μM GTP; (**B**) Line profile of height displacement along the red-line in (**A**); (**C**) Filamentous structure of SaFtsZ in the crystal (PDB entry: 3VOA).

**Figure 2 ijms-22-01697-f002:**
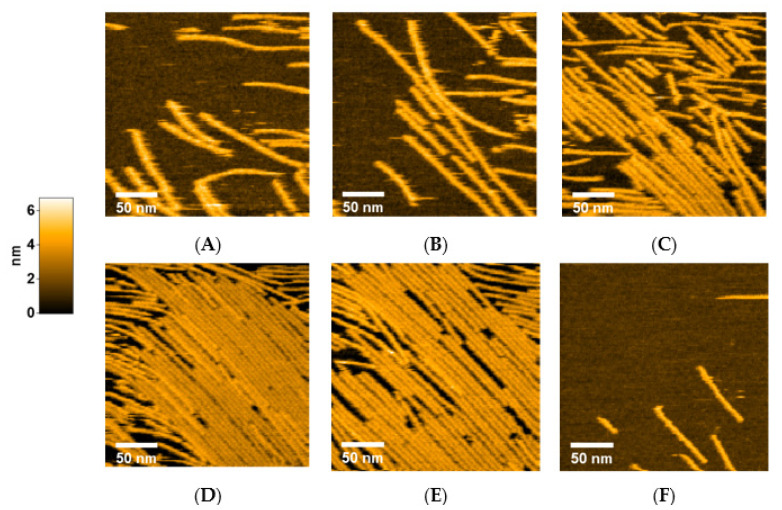
Growth and dissociation of FtsZt filaments. Each image was taken under the following conditions: (**A**) 100 nM FtsZt and 300 μM GTP; (**B**) 100 nM FtsZt and 600 μM GTP; (**C**) 200 nM FtsZt and 600 μM GTP; (**D**) After 20 min incubation from (**C**); (**E**) 200 nM FtsZt, 600 μM GTP, and 3 mM GDP; (**F**) 200 nM FtsZt, 600 μM GTP, and 6 mM GDP. Scale bar: 50 nm.

**Figure 3 ijms-22-01697-f003:**
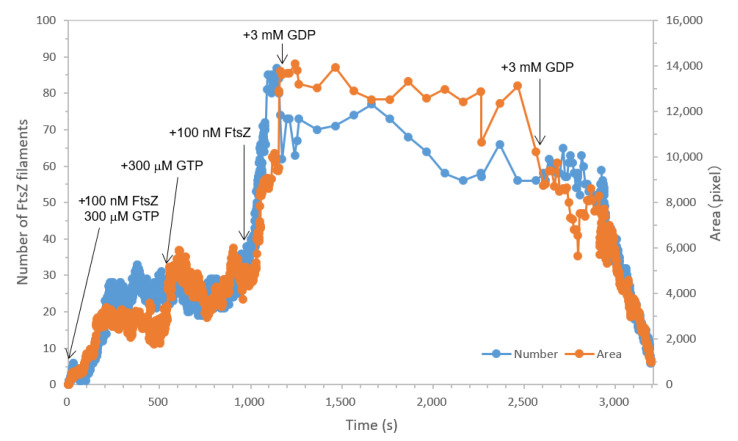
Time-course graph of the number (blue) of FtsZ filaments and the area (orange) covered with FtsZ filaments shown in Figure 2. Note that the time course is not completely continuous because of the intervals for adding samples and changing settings.

**Figure 4 ijms-22-01697-f004:**
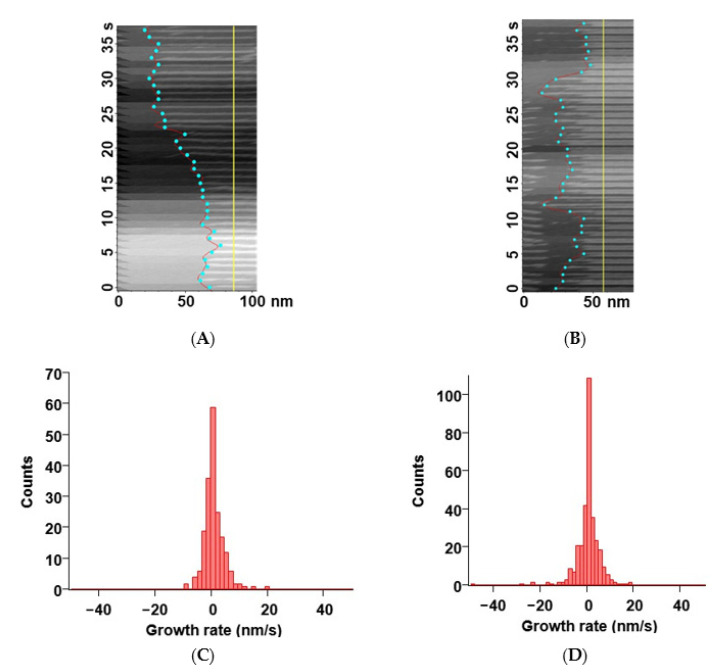
Growth and dissociation rates of FtsZt filaments: (**A**,**B**) Typical kymographs of a single FtsZt filament in the growth condition (**A**) and the dissociation condition (**B**). Cyan circles represent the position of filament ends on each frame. Yellow line shows the base line for the measurement of length. The time axis goes from the bottom to the top; (**C**,**D**) Histograms of growth/dissociation rates of FtsZt filaments in the growth condition (**C**) and the dissociation condition (**D**). Scale bars: 50–100 nm.

**Figure 5 ijms-22-01697-f005:**
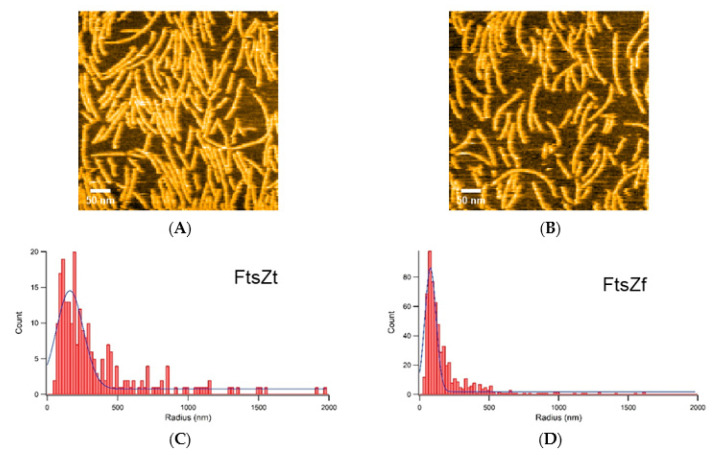
Curvature analysis of FtsZt and FtsZf filaments: (**A**,**B**) An image of FtsZf filaments taken under the following conditions: (**A**) 100 nM FtsZf and 600 μM GTP; (**B**) 100 nM FtsZf, 600 μM GTP, and 6 mM GDP; (**A**,**B**) show different fields of view; (**C**,**D**) Histograms of radius calculated by regarding the FtsZt (**C**) and FtsZf (**D**) filaments as arcs. Scale bars: 50 nm.

## Data Availability

Data is contained within the article.

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
