# Peer review of "Dynamic Assembly/Disassembly of Staphylococcus aureus FtsZ Visualized by High-Speed Atomic Force Microscopy"

_ijms, 2021, doi:10.3390/ijms22041697_

Round 1

Reviewer 1 Report

I consider that the manuscript has been significantly improved and that the authors have addressed and responded to my concerns. 

 I think it can be published as it is in this new version.

Reviewer 2 Report

I really see and appreciate the effort to improve the manuscript.

My only concern is that movie S5 seems as two experiments joined in a single movie (one from second 1 to 9 and, second from second 10 to 27) but there is no clarification about this point in the manuscript.

This manuscript is a resubmission of an earlier submission. The following is a list of the peer review reports and author responses from that submission.

Round 1

Reviewer 1 Report

The manuscript by Fujita et al presents the assembly dynamics of bacterial cell division FtsZ using High-speed AFM achieving sub-second time resolution. To this end, they use FtsZ from Staphylococcus aureus. They build two different constructs: the full-length and a truncated version that lacks the N-termini and C-termini flexible tags. Their starting point in all the experiments is an apo-protein (no nucleotide bound at the nucleotide site) and performed their experiments in a common FtsZ buffer, which could easily correlate their data with other in vitro approaches.

I think authors should take care with their claims. For instance, they mention that disassociation of FtsZ filaments have not been observed previously. However, Mateos-Gil P et al. PNAS 2012 already described depolymerization dynamics of individual FtsZ filaments by AFM.

I have found an interesting manuscript with valuable data (filament treadmilling, filament fusion/repulsion, lateral contacts formation). But I think these data need further analysis to fully appreciate the specific claims of the manuscript. Particularly:

  1. Polarity of the filaments is determined due to differences on each end growing speed, but authors do not contribute with any data to this claim. They should do a proper measurement of the assembly speed (i.e. considering a statistically significant number of filaments ends to get also an estimation of the error in their measurements).
  2. Straight-vs-Curved filaments. There are many papers showing FtsZ curved protofilaments and, it has been proposed that lateral contacts contribute to filament straightening. Instead, authors insist on their images show mainly straight filaments. Please, you should notice that most of your images comes from SaFtsZt, which may contribute on filaments straightening. In fact, Fig. 5 show experiments with SaFtsZf, and curved filaments are clear considering the small visualization field displayed (which I guess is necessary to reach high-speed measurements but otherwise is definitely much smaller than fields showed in other AFM studies). Bigger visualization fields would give a rough idea of FtsZ filaments curvature. Additionally, previous AFM studies showed that decreasing the total amount of FtsZ contributed to decrease significantly straight filaments vs. curved filaments (denoting that lateral contacts may be involve in filaments straightening). Therefore, filaments straight conformation should be considered within the limitations of the technique and the approach used.
  3. Bending filaments during depolymerization. According to the videos presented depolymerization is very fast and difficult to determine the curvature of the end of the filaments at each frame. However, authors should improve this section with a detail analysis of filaments depolymerization (i.e. measure the depolymerization rate within their experiments considering a statically significant number of filaments). From the videos presented the depolymerization is very fast, but determination of filament length frame-by-frame could give an approximation to the depolymerization rate.
  4. Lateral contacts: There are several papers pointing to the C-terminal tail as an unstructured element directly involved in lateral association of FtsZ filaments. However, they see such interactions even in the SaFtsZt. Considering that they also perform experiments with SaFtsZf, they should analyze lateral interactions (similarly to the analysis performed on axial interactions in Fig. 1) and present if there are any differences on the lateral distance between filaments.

Finally, related to their comment on "some further X-ray crystallography to examine detailed molecular mechanism in filament assembly and depolymerization" I would like to highlight the following: considering that there are several high-resolution structures at the Protein Data Bank covering two main conformations (T and R) and a wide range of nucleotide bound states of SaFtsZ, I feel that the field of FtsZ structural studies needs some insight from cryoEM studies on FtsZ filaments rather than more X-ray crystal structures.

Reviewer 2 Report

The work “Dynamic Assembly/Disassembly of Staphylococcus  aureus FtsZ Visualized by High-Speed Atomic Force   Microscopy “ by Junso Fujita  , Shogo Sugiyama  , Haruna Terakado  , Maho Miyazaki  , Mayuki Ozawa  ,   Nanami Ueda  , Shun-ichi Tanaka  , Takuya Yoshizawa , Takayuki Uchihashi , and  Hiroyoshi Matsumura presents very nice data observing the polymerization and depolymerization dynamics of Staphylococcus aureus FtsZ visualized by high-speed AFM. Previous work characterizing polymerization depolymerization with AFM was done using E. coli FtsZ, so it is interesting to observe the behavior of FtsZ from another organism and using high-speed AFM, accessing faster rearrangement times. However, I don´t think that the  interpretation of the results is consistent with the observations, and I think that the amount of new information obtained from these experiments should be more thoroughly discussed, as I will explain shortly.

1) One first comment is that it would have been interesting to observe the behavior of the full-length protein. Using a high speed AFM provides a better time resolution than a regular AFM, but the process observed and the dynamic behavior on the surface are qualitatively the same. Previous work was done with a regular AFM. This could actually serve as a good reference to see what additional information can be obtained with the fast AFM. In previous work ( ref 22 in the paper) changing the pH of the solution slowed down depolymerization dynamics until it could be followed by AFM.

2) Another comment refers to the fact that, after truncating the protein, the majority of the filaments observed are straight. Again, it would be very interesting to compare the behavior of the full length protein. It is not clear why on figure 5 the authors do show the full length protein. Filaments appear to be more curved, but it is difficult to see from this image that presents only a small area.

If the authors looked carefully at reference 21 and also at ref. FEMS Microbiology Reviews, fuy039, 43, 2019, 73–87 doi: 10.1093/femsre/fuy039 (not cited in the paper), one possible explanation for the observation of mostly straight filaments could be that removing the terminal region of the protein affects the way they interact with the mica, favoring the observation of straight filaments instead of curved ones.

3)   I disagree with the interpretation of the authors that their results show a clear polarity in filament growth. Although a treadmilling  behavior of the filaments is observed in cells, using single molecule fluorescence techniques ( which means growing from one end and shortening from the other), the data obtained from isolated filaments on mica and on lipid surfaces (ref 22 and Int. J. Mol. Sci. 2019, 20, 2545; doi:10.3390/ijms20102545, also not mentioned in the paper) do not show a clear treadmilling behavior.  Growth and reduction are random and they can alternate on both ends. One may start growing on one direction, stop, and start growing in the other direction, or both ends can grow. This is what is seen on many filaments in the images and the videos presented by the authors.  There are several examples of these more random behaviors in the movies presented in the supplementary material, particularly in movieS3 in which you can analyze several individual filaments when they become less dense, as they still grow before the fully depolymerize. The behavior appears to be random, very similar to what was describes in ref 22 for E. coli FtsZ.  It would be interesting to do a statistical analysis of the behavior of the ends of several individual filaments in order to obtain a clear answer. One possible explanation in order to reconcile the treadmilling behavior observed in vivo with the more random growth observed for individual filaments could be the explanation provided in ref. FEMS Microbiology Reviews, fuy039, 43, 2019, 73–87 doi: 10.1093/femsre/fuy039, that argues that the apparent treadmilling behavior could be a consequence of the type of attachment of the filaments to the surface.

In conclusion, I  consider that the conclusions drawn are not convincingly supported by the data presented, so the paper cannot be published in its present form.
